# Physical and Chemical Effects in Blended Cement Pastes Elaborated with Calcined Clay and Nanosilica

**DOI:** 10.3390/ma16051837

**Published:** 2023-02-23

**Authors:** Divino Gabriel Lima Pinheiro, Matheus Ian Castro Sousa, Fernando Pelisser, João Henrique da Silva Rêgo, Amparo Moragues Terrades, Moisés Frías Rojas

**Affiliations:** 1Department of Civil and Environmental Engineering, University of Brasília (UnB), Brasília 70910-900, Brazil; 2Department of Civil Engineering, Laboratory of Application of Nanotechnology in Civil Construction (LabNANOTEC), Federal University of Santa Catarina (UFSC), Florianópolis 88040-900, Brazil; 3Department of Civil Engineering: Construction, Polytechnic University of Madri, Calle del Prof. Araguren, 3, 28040 Madrid, Spain; 4Eduardo Torroja Institute (CSIC), Spanish National Research Council, 28033 Madrid, Spain

**Keywords:** calcined clay, nanosilica, ternary cements, hydration kinetics, microstructure

## Abstract

Supplementary cementitious materials (SCMs) are commonly used in the manufacture of commercial cements with lower clinker content and carbon footprints, enabling environmental and performance improvements. The present article evaluated a ternary cement combining 23% calcined clay (CC) and 2% nanosilica (NS) to replace 25% of the Ordinary Portland Cement (OPC) content. For this purpose, a series of tests were performed, such as compressive strength, isothermal calorimetry, thermogravimetry (TG/DTG), X-ray diffraction (XDR), and mercury intrusion porosimetry (MIP). The ternary cement studied, 23CC2NS, presents a very high surface area, which influences hydration kinetics by accelerating silicate formation and causes an undersulfated condition. The pozzolanic reaction is potentialized by the synergy between the CC and NS, resulting in a lower portlandite content at 28 days in the 23CC2NS paste (6%) compared with the 25CC paste (12%) and 2NS paste (13%). A significant reduction in total porosity and conversion of macropores in mesopores was observed. For example, 70% of pores in OPC paste were macropores that were converted in the 23CC2NS paste into mesopores and gel pores.

## 1. Introduction

The use of supplementary cementitious materials (SCMs) to form blended cements as a strategy to reduce clinker content has increased over the years in the cement industry, diminishing the average global clinker content from 85% in 2003 to 71% in the near future [1,2,3,4,5,6,7,8,9,10]. The most common SCMs used with this aim are blast furnace slag and fly ash, but limitations in their availability create a demand for new alternatives, such as calcined clay.

Cement production is responsible for up to 7% of global CO_2_ emissions into the atmosphere. To produce clinker, with burning temperatures around 1400 °C, the process emits approximately 850 kg/CO_2_.ton. One of the objectives of using calcined clay as an SCM is to reduce CO_2_ emissions and energy demand in the production process. CO_2_ emissions in the production of calcined clay are largely derived from the burning of fuels (85% of the total). Even so, the calcined clay production process, with burning between 600 and 950 °C, emits only 270 kg/CO_2_.ton [11,12,13,14,15].

Calcined clays (CCs) are produced by the calcination of kaolinitic-based clays at 550–900 °C, resulting in the conversion of kaolinite into metakaolinite, a highly reactive pozzolanic component [12,16,17,18,19]. Clay deposits suitable to this application are common worldwide, facilitating their use in the cement industry, but their use is still limited to a few countries [1,20,21,22,23]. Therefore, research regarding the use of CC in blended cements has increased, both in binary mixtures with OPC and in ternary mixtures, such as LC^3^ cement [24,25,26,27]. 

When used as an SCM, the main influence of CC occurs via the pozzolanic reaction of metakaolinite with Portlandite (CH) to form additional C-S-H (hydrated calcium silicate), C-A-S-H (hydrated calcium aluminosilicate) such as C_2_ASH_8_ (stratlingite), and aluminate hydrates such as C_4_AH_12_ [28,29,30]. This reaction tends to occur more significantly after 7 days of hydration, contributing to compressive strength at later ages. Furthermore, CC also affects cement hydration through physical effects, such as nucleation, when the particles have small dimensions, and dilution, since there is more space available for hydration reactions due to the lower clinker content [31].

Considering the slow start in CC pozzolanic reaction, where better results are obtained after 28 days, an interesting possibility for ternary mixtures is the use of nanosilica (NS). This SCM is highly reactive due to its nanometric dimensions, high specific surface area, and high content of amorphous SiO_2_ (above 90%) [32]. Its physical properties promote the acceleration of the hydration reaction, mainly due to a potentialized nucleation effect, and chemically it reacts with CH to form additional C-S-H as early as 1 day in the hydration process. As a result of these effects, NS is known to drastically improve early age compressive strength, as well as refine porosity and densify the microstructure of the cementitious matrix [33,34].

Previous studies [35,36,37,38,39,40] already showed the potential of ternary cement pastes with nanosilica and metakaolinite (MK), having a synergistic effect between the materials. Mixtures with NS and MK showed lower porosity, higher compressive strength values, and C-A-S-H formed with higher aluminum content and chain length. 

Although there are already studies on NS combined with MK, and also studies on CC (isolated or combined with limestone [41,42,43,44]), there is a lack of publications on ternary cements with CC and NS. Such blended cement could show improved characteristics considering both sustainability and the interactions between its components. Therefore, the present work aimed to study such pozzolan mixture, evaluating the interaction between these SCM in cement pastes containing NS, CC, or inert filler, both in binary and ternary arrangements. Inert filler was used to distinguish chemical and physical effects between the materials.

## 2. Materials and Methods

### 2.1. Materials

The following materials were used in this work: Portland cement CPI-40 (OPC), produced by Votorantim Cimentos, classified according to the Brazilian standard [45]; Calcined clay (CC), produced in the laboratory by calcining of a kaolinitic clay used in calcined clay production in cement industry; Inert siliceous filler (IF), produced in the laboratory by grinding Brazilian standardized sand [46]; Nanosilica (NS), consisting of a colloidal solution of SiO_2_ nanoparticles (30% weight) and water (70% weight), produced by AkzoNobel; Superplasticizer admixture ViscoCrete 6900, based on a polycarboxylate solution with 46% solid content, light brown liquid aspect, PH 5.5 ± 1.0, density 1.1 ± 0.02 kg/L, which acts through the effects of superficial adsorption and steric separation in the cement particles, produced by Sika.

The calcination of the raw clay was performed at 850 °C for 2 h in a muffle furnace. The temperature was chosen according to previous findings in the literature regarding ideal calcination temperatures for CC [47,48,49]. After calcination, the clay was cooled naturally until reaching room temperature.

Grinding was performed using a Los Angeles abrasion test machine with 15 steel balls for both CC and IF. The grinding time was adjusted based on a pilot study, testing various grinding times to CC and IF and by taking the granulometry of OPC as a reference. The final grinding time for CC and IF was 4 and 8 h, respectively.

The physical and chemical properties of the materials are shown in Table 1. Particle size distribution parameters were obtained by laser granulometry for OPC, CC, and IF, and by dynamic light scattering for NS. Diffractograms were obtained by X-Ray diffraction (XRD), which was performed using a Bruker D8 Discover diffractometer, manufactured by the company Bruker (Karlsruhe and Madison, Wisconsin, USA), for the individual materials. Specific surface area (SSA) was obtained by the B.E.T. method except for NS. Specific mass was determined with a gas pycnometer for all materials except NS. For both properties, values for NS were informed by the manufacturer. Finally, chemical composition was obtained by X-ray Fluorescence Spectroscopy, performed after a loss on ignition test. Before the loss on ignition test, NS was air dried for 72 h to analyze only the solid material.

The chemical composition of the OPC met the requirements of the Brazilian standard, which stipulates the maximum amounts for LOI, MgO, SO_3,_ and insoluble residue. CC composition included high values of SiO_2_ and AlO_3_, which surpassed 90% when combined. NS consisted of mainly SiO_2_, with a low content of Na_2_O. It is worth noting that the loss on ignition value was relatively high, but this was probably moisture that was not fully removed by the air-drying process. Finally, regarding IF, the chemical composition results show mostly SiO_2_ and low contents of impurities. Since this SiO_2_ is related to quartz, the result is important to confirm the consideration that the material will be inert.

Particle size parameters show that both CC and IF were close to OPC in terms of particle size distribution, which was the aim of the grinding process. It is worth noting, however, that the SSA of CC was considerably higher than that of OPC and IF.

The metakaolinite content of CC was also evaluated since it is the main reactive component of this SCM. The Kaolinite content of the raw clay and CC were calculated using the weight loss of the dehydroxylation of kaolinite that occurs between 400 °C and 600 °C [37], as shown in Figure 1. The weight loss was obtained through Thermogravimetric Analysis (TGA). The values found were 54% of kaolinite for the raw clay and 3.1% for the CC. The difference between those samples is the kaolinite content that was converted into metakaolinite. Therefore, the CC studied has a metakaolinite content of 50.9%.

X-ray diffractograms show the mineralogy of the materials. It can be seen that IF, Figure 2a, has a diffractogram with no amorphous halo and with quartz as the most evident peak, confirming the consideration that the material is inert. In CC, Figure 2b, kaolinite and quartz peaks are detected, and an amorphous halo can be observed between 15° and 30° [50]. The kaolinite peak refers to the remaining kaolinite content after calcination, measured by thermogravimetric analysis (3.1%), while the amorphous halo most likely represents the metakaolinite formed in calcination. For OPC, Figure 2c, the main phases alite (Ca_3_SiO_5_), belite (Ca_2_SiO_4_), tricalcium aluminate (C_3_A), and periclase (MgO), ferrite (Ca_2_Fe_2_O_5_) were identified. NS, Figure 2d, shows a completely amorphous structure, with no crystals detected, which is consistent with its composition (amorphous particles of SiO_2_).

### 2.2. Past Composition and Preparation

Six different cement pastes were prepared, one for reference containing only OPC and five blended cement pastes. In one of the blended pastes, 2% of OPC was replaced by NS forming a binary mixture, while in the other four, 25% of OPC was replaced by the SCMs forming binary and ternary pastes. NS content was chosen considering previous research [35,38], while the 25% replacement content was chosen considering the Brazilian standard (ABNT 2015b). The water/binder ratio was fixed at 0.40 for all pastes. Consistency was measured by the mini-slump test and the parameter was fixed at 94 ± 4 mm, considered ideal for molding [51]. This consistency was achieved using a superplasticizer admixture. It is worth noting that the water content of NS and the superplasticizer were discounted from the water used in the mixture to ensure that all pastes had the same water content. Table 2 shows the detailed paste compositions.

Regarding the consistency of the pastes, a general behavior of increased demand of superplasticizer admixture to reach the same consistency in pastes with materials of greater surface area such as NS and CC was observed, which is consistent with other references [39,52,53,54]. This occurs because more water is needed to cover the particles as the surface area of the materials increases, thus leaving less free water and decreasing the consistency [55,56]. The difference between the pastes with CC and IF is explained by this behavior. Both have similar dimensions and were used with the same replacement content, but as the surface area of CC is much higher, the demand for superplasticizers was considerably different. 

The pastes were prepared in a climatized room with a temperature of 23 ± 1 °C, according to the Brazilian standard [46]. Mixing was performed in the following order: first, a mix of the liquids (water, superplasticizer admixture, and NS) was added to the mixer, followed by the addition of cement/SCM; later, the planetary mixer was used for 60 s at a slow rotation, followed by 90 s of rest and then 90 s in the fast rotation of the mixer. Molding was performed using cylindric molds of 5 cm × 10 cm, which were placed in a humid chamber for 1 day before demolding. Then, the pastes were cured by immersion in lime saturated water until the testing age.

The test specimens were used for the compressive strength test. For other tests, except mercury intrusion porosimetry (MIP), samples were obtained from fragments of the ruptured test specimen, discarding border regions, and choosing only from the center area. For the MIP test, cubic samples with approximately 2 cm dimensions were obtained from non-ruptured specimens using a circular saw. 

Hydration was stopped in the samples by immersion in isopropanol for 24 h and drying at 40 ± 1 °C for 24 h. This protocol was adapted considering recommendations in the literature [57]. After this process, the samples were placed with silica gel and soda lime in containers that were sealed until the testing day.

### 2.3. Test Methods

Isothermal calorimetry was carried out using a Thermometric TAM Air calorimeter from TA Instruments, with 8 channels and temperature control. Data were collected by the PicoLog software. 

X-Ray diffraction (XRD) was performed using a Bruker D8 Discover diffractometer, with monochromatic radiation from a tube with a copper anode and using a Johansson monochromator for Kα1, operating at 40 kV and 40 mA, Bragg–Brentano θ-2θ configuration, using a unidimensional Lynxeye detector. Measurements were carried out between 5 and 25° 2θ, with 0.01° steps and 15 rpm rotation during the test. 

The granulometry by laser beam was carried out in a Cilas Particle Size 1180 equipment, capable of providing a particle size distribution between 0.04 μm and 2500 μm in dimension. The dispersing medium was alcohol (99.8%).

Thermogravimetric Analysis (TGA) was performed in a Shimadzu DTG-60H, between 25 e 600 °C, with a heat ramp of 20 °C/min, nitrogen flow of 10 mL/min, using aluminum crucibles of 70µL. The sample mass is approximately 15 mg.

Mercury intrusion Porosimetry (MIP) was carried out using a Micromeritics Poresizer, model 9320. The contact angle was 130 °C; mercury with surface tension of 0.485 N/m and specific mass of 13.5335 g/mL was used. The pressure interval used in the test was between 0.50 psi and 29,472.38 psi.

Compressive strength tests were performed following the Brazilian standard [46], with three test specimens ruptured at 1, 3, 7, and 28 days. 

## 3. Results

### 3.1. Isothermal Calorimetry

The progress of hydration in the pastes was evaluated by isothermal calorimetry. Figure 3a shows their overall heat of hydration, while Figure 3b shows the heat flow curve, regularized by clinker content in the pastes. Some parameters obtained by analyzing the curves presented in Figure 3 are presented in Table 3.

The regularization by clinker content used in the heat flow curve shows how the presence of SCMs influenced the behavior of clinker hydration. Overall, the main hydration peak of cement happened earlier and with greater heat flow value in pastes with SCM, when compared with OPC paste, which is consistent with the literature [28,37,38,40]. In the ternary paste, 23CC2NS, the main peak happened 4.4 h earlier than in the OPC paste. 

In pastes containing CC or NS, the anticipation of the main peak was aided by the steeper inclination of the curves in the acceleration period, which is explained by the nucleation effect [28,35,58,59,60]. This effect is more noticeable in pastes with NS, which is expected due to its high surface area in addition to the pozzolanic effect that already occurs in the first hours of hydration. Regarding the increase in heat flow intensity at the main peak in pastes with SCM, part of the behavior is explained by the dilution effect, since the clinker has more available space and thus can react more freely [61,62].

The shape of the curves also varied between pastes when considering the main peak width and the position of the bump in the curves related to the aluminate peak. While OPC and 25IF pastes had a similar shape overall, with the bump occurring after a maximum in the acceleration period and presenting similar width, the remaining pastes had a different behavior. Pastes with NS and/or CC had the bump in the curve occurring before the maximum point that marks the end of the acceleration period, and the result observed in the curve was a sudden change in steepness. For example, in paste 25CC this change can be seen at 6 h. 

Such behavior in the hydration of blended cement pastes has been linked to the sulfate balance during hydration, and the change occurs due to the difference in surface area between the materials [63]. In pastes containing SCM with higher surface area, the nucleation effect accelerates the precipitation rate of C-S-H and more sulfate is adsorbed by it, shifting the aluminum peak since it is linked to sulfate depletion. 

It is worth noting that this undersulfation condition, characterized by the superposition of the main silicate and aluminate peaks, has detrimental effects on main peak hydration, which later on can negatively impact overall strength in cementitious materials [64,65,66,67]. This effect can help explain the behavior of the overall heat of hydration observed in the pastes. When comparing the pair 25IF/23IF2NS with the pair 25CC/23CC2NS, the heat of hydration at 24 and 72 h is higher in the former, which would not be expected considering the acceleration effect due to higher surface area and the fact that they all have the same clinker content. However, by taking the negative effects of undersulfation condition into account, it can be inferred that in the pastes with CC, the hydrate formation was hindered.

### 3.2. X-ray Diffraction (XRD)

XRD patterns obtained for all pastes at 1, 3, 7, and 28 days of hydration are shown in Figure 4. The main minerals identified were Portlandite, Ettringite, Monosulfoaluminate, Alite, Belite, Periclase, Calcite, and Quartz. The quartz peak located at 22–23° was present in pastes with CC, and IF, showing that both materials contain the mineral. In the case of the IF, the peaks are significantly higher and help confirm the assumption that the silicious filler chosen is inert. Peak intensity was constant at all ages, as expected of quartz as an inert material. 

Ettringite peaks at 9° were detected in all pastes, with higher intensity at earlier ages. As hydration progresses, ettringite is converted to monosulfoaluminate with the peak located at 10°. At 28 days, the ettringite peak is only detected in the OPC paste. 

The Portlandite peak, at 17–18°, was present in all pastes at all ages. Pastes with NS have notably smaller peaks when compared with their counterparts without NS at each age, even 1 day. This is a direct result of the pozzolanic reaction, responsible for consuming CH [68,69]. Furthermore, pastes with CC have lower peak intensity than both OPC and counterpart pastes with IF. The comparison between CC and IF pastes shows that the lower peak intensity is not just a consequence of lower clinker content, but also a result of a chemical effect, the pozzolanic reaction. The 23CC 2NS paste had the lowest peak intensity at all ages, consistent with the literature due to the combination of CH-consuming effects [37]. The periclase peak, at 42° in the pastes, is justified by the amount of MgO present in the OPC sample.

### 3.3. Thermogravimetric Analysis (TGA) 

TGA was used to quantify material content in the pastes using the mass loss at specific intervals. The test was performed at 1, 3, 7, and 28 days of hydration for all pastes. Figure 5 shows the TGA curves for all pastes at 28 days.

The decomposition of CH was evaluated by determining the initial and final temperatures with a graphical analysis of the curves in the region of approximately 400 °C and 500 °C. CH content was then calculated by multiplying the mass loss by 4.11 (molar mass of CH/molar mass of water). CH content over time is shown in Figure 6.

Overall, CH content decreased in the following order: OPC, 25IF, 2NS, 25CC, 23IF2NS, and 23CC2NS. CH content depends on clinker content, which determines the amount of CH that can be produced during hydration; hydration kinetics, which regulates that production over time; and pozzolanic reaction, which is responsible for consuming available CH. The behavior of the materials can be understood when considering all these effects. 

The presence of NS promotes pozzolanic reaction from a very early age, resulting in CH consumption detectable at 1 day. This can be concluded since the clinker content in paired pastes with and without NS is roughly the same and the hydration rate of clinker is higher (higher heat of hydration measured with isothermal calorimetry), which would lead to more CH produced. Since the CH content was lower in these conditions, the only explanation is pozzolanic activity.

The pozzolanic reaction of CC starts later than that of NS. Comparing pastes containing a reactive and an inert component, it was noted that the CH content in pastes 25CC and 25IF was very similar at 1 day of hydration. Since both have similar clinker content and the hydration rate is not higher for CC paste, it can be concluded that there was no significant CH consumption due to pozzolanic activity by CC. The reason both have lower CH content than OPC paste at this age is simply due to lower clinker content. From 3 days onwards, the same comparison yields a different result, with the 25CC paste always having lower CH content than the 25IF paste. Therefore, it can be concluded that pozzolanic activity for the CC used starts at 3 days.

Regarding the ternary cement, it had the lowest CH content at all ages. Besides the combination of the aforementioned causes, it is also worth noting that the gap in CH content caused by NS when comparing pastes 25CC and 23CC2NS was higher than the one between pastes 25IF and 23IF2NS. At 1 day, NS reduced the CH content by 41% when comparing both pastes with CC, but it only reduced the CH content by 15% when comparing both pastes with IF. This behavior was repeated in the following ages, although the difference decreased. This result shows that there is a synergistic interaction between CC and NS, supporting previously reported results [35,38]. 

Furthermore, the hydrated phases content was also evaluated at 28 days using the mass loss at the interval 50–440 °C associated with the dehydroxylation or dehydration of the major mineralogical phases generated during Portland cement hydration or pozzolanic reaction. In this temperature range, there is the dehydration of the C-S-H gel, loss of water from ettringite, dehydroxylation of tetracalcium aluminate hydrate, and also the dehydroxylation of hydrated monocarboaluminate [14,70,71]. Although not representative, the value of the C-S-H content in the paste was used to calculate a Hydrated Phases index (HP index) for the pastes, which directly compares them with the OPC paste. Table 4 compiles the results.

The paste 25CC showed an increase in the HP index compared with the OPC, and the increase was even higher in the paste 23CC2NS [37]. The highest HP index was found in the 23CC2NS paste, in which the C-S-H produced by the pozzolanic reaction was added to that formed by the hydration of the clinker. The pastes with IF had an HP index lower than 100%. In the 23CC2NS paste, the synergy between CC and NS is observed. In this case, the hydrated phases produced by the combination of OPC hydration and the pozzolanic reaction of the other materials used (CC and NS) outweigh the hydrated phases of the OPC paste. The OPC paste contains more clinker than other pastes because in the other pastes part of the OPC cement has been replaced by SCMs. 

### 3.4. Mercury Intrusion Porosimetry (MIP)

MIP tests were carried out to evaluate the pore structure of the cementitious matrix at 28 days of hydration. Pore size distribution graphs are shown in Figure 7, with separating regions of gel pores (lower than 10 nm), mesopores (between 10 and 50 nm), and macropores (higher than 50 nm) [72,73].

Figure 7a shows that there was a refinement of the pore structure in pastes containing CC and/or NS, with a shift of the intrusion regions in direction of lower pore diameters. This was particularly noteworthy in paste 23CC2NS. In those pastes, besides the filler effect that occurs with finer particles occupying spaces between the cement grains, the pozzolanic reaction produces additional C-S-H that further fills pore spaces [74,75,76].

Figure 7b shows the accumulated intrusion, and the maximum value is related to the total porosity in the sample. Paste 25IF has the highest total porosity, as expected due to lower clinker content and thus lower hydrate production. In paste 23IF2NS, although there was a shift to lower pore sizes, the total volume still surpassed the OPC paste. Pastes with CC show the lowest total pore volume and pore refinement, observed in Figure 7a, resulting in the most compact and dense microstructures [60,77].

To better understand the refinement that occurs in pastes with SCM, the intrusion volume in each region was calculated concerning the total intrusion volume. The distribution between the regions is shown in Figure 8. In the OPC paste, more than 70% of the total intrusion volume occurs in macropores. In the other pastes, the macropore fraction was reduced and converted into mesopore and gel pore fractions. In paste 23CC2NS, almost none of the macropore region remains. Compared with OPC, even the paste 25IF had a small refinement (fewer macropores), probably due to the filler. 

### 3.5. Compressive Strength

Compressive strength values were obtained for the pastes at 1, 3, 7, and 28 days of hydration and are presented in graph form in Figure 9. 

Pastes with NS showed higher values than their counterparts at all ages, but the difference between them decreased over time. At the early ages, the acceleration in hydration caused by the nucleation effect, which was observed in Isothermal Calorimetry data, significantly increases compressive strength. When comparing OPC and 2NS paste, the difference found was 82% at 1 day, 41% at 3 days, and 35% at 7 days. Over time, however, hydrate formation manages to catch up, resulting in a difference of only 5% between pastes OPC and 2NS. This behavior is consistent with the literature regarding NS [78,79,80].

The paste with IF acted as expected, following a strength gain pattern similar to OPC, but always at lower values, due to the deficit in clinker content. Until 7 days, all combinations with NS have higher compressive strength than the OPC, when the effect of acceleration is more significant.

The impact on the compressive strength of CC, in both binary and ternary mixture, was lower than expected, considering previous works [63,64], particularly in the case of paste 23CC2NS. Interaction between NS and materials containing metakaolinite showed synergistic effects in previous studies, resulting in compressive strength gain at later ages [51,57,65]. In this study, however, the difference between the 23CC2NS paste and the 25CC paste was only observed at early ages. One possible explanation is the detrimental effect of the undersulfated condition, which limits reactivity and hinders hydrate formation, observed with isothermal calorimetry data. Increased aluminum concentrations in solution as a result of initial C3A hydration in a subsulfated cement may negatively influence C3S hydration [64,65,66,67] and, consequently, may decrease the compressive strength of the cementitious material. Despite this, as shown in Figure 9, a tendency towards an increase in the compressive strength of the 25CC and 23CC2NS pastes was observed, which will probably lead them to exceed the compressive strength of the 2NS paste after 28 days of hydration.

## 4. Conclusions

Considering the analysis of the data obtained with experimental tests, the following conclusions were drawn: Hydration kinetics were heavily influenced by the presence of NS and CC, with acceleration due to nucleation effects in both binary and ternary. In these mixtures, which possess higher total surface area and lower clinker content than OPC, an undersulfated condition was observed, with superposition of silicate and aluminate peaks;Analysis of CH content in the pastes over time shows that the pozzolanic re-action starts before 1 day for NS, and between 1 and 3 days for CC. Furthermore, the effect of NS in CH consumption was more significant in paste 23CC2NS than in paste 23IF2NS at all ages, reaffirming the chemical effect of the pozzolanic reaction of CC when compared with IF;The use of NS and CC, in both binary and ternary mixtures, promoted a reduction in total porosity and refinement of the microstructure, with the conversion of macropores into mesopores and gel pores. This effect was very significant in the ternary paste (23CC2NS), which showed macropore volume close to zero;Compressive strength was increased at an early age with the use of NS. At later ages, however, all pastes without IF showed similar compressive strength values. In the case of 23CC2NS paste, although a higher compressive strength was expected, the undersulfated condition might have limited the reactive potential of the materials and consequently the potential for compressive strength gains.

## Figures and Tables

**Figure 1 materials-16-01837-f001:**
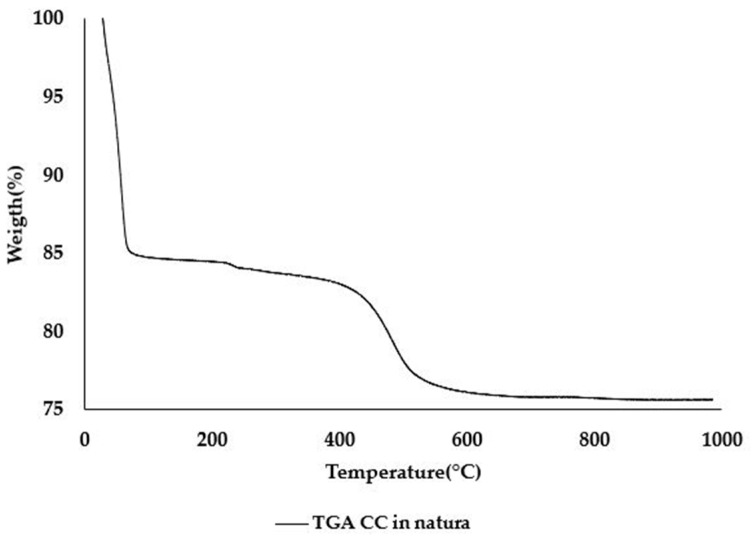
Natural CC TGA curve.

**Figure 2 materials-16-01837-f002:**
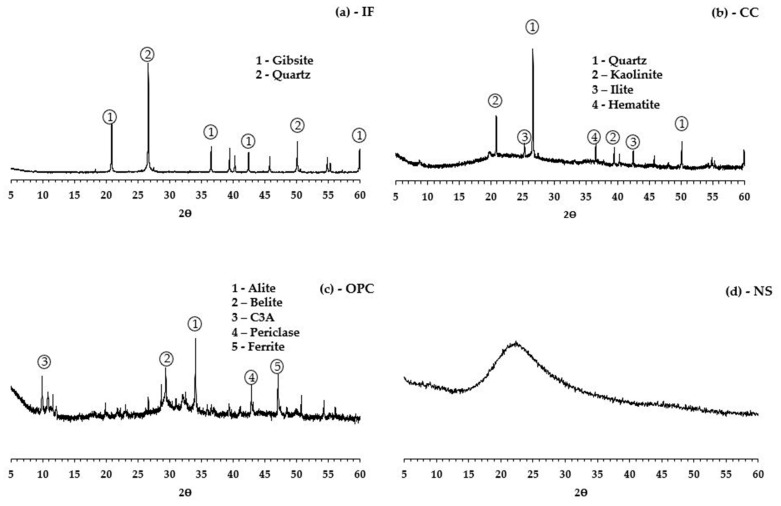
(**a**) XRD IF, (**b**) XRD CC, (**c**) XRD OPC, and (**d**) XRD NS.

**Figure 3 materials-16-01837-f003:**
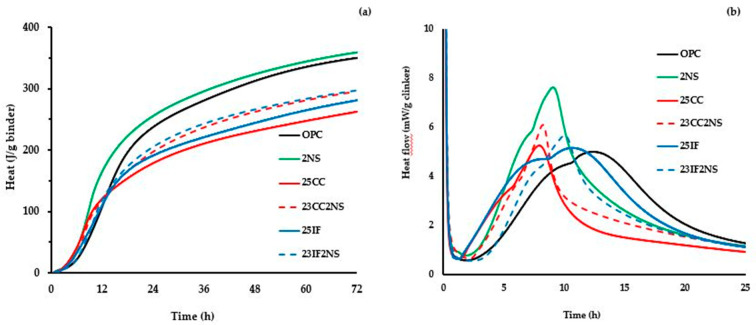
(**a**) Heat of hydration of pastes regularized by binder content until 72 h of hydration. (**b**) Heat flow of pastes regularized by clinker until 24 h of hydration.

**Figure 4 materials-16-01837-f004:**
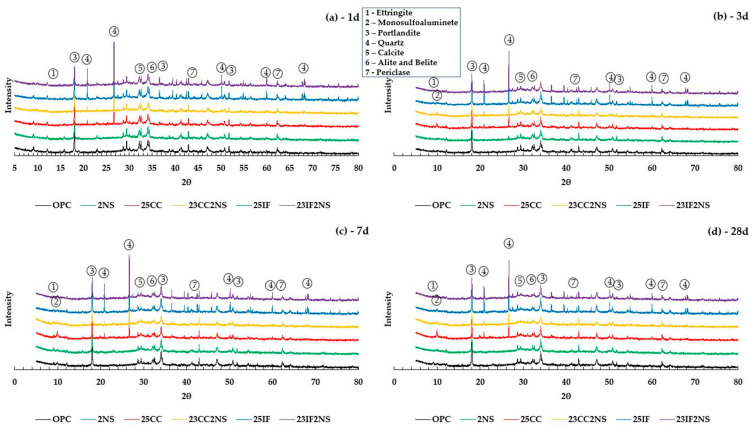
XRD patterns obtained for all pastes at (**a**) 1 day, (**b**) 3 days, (**c**) 7 days, and (**d**) 28 days of hydration.

**Figure 5 materials-16-01837-f005:**
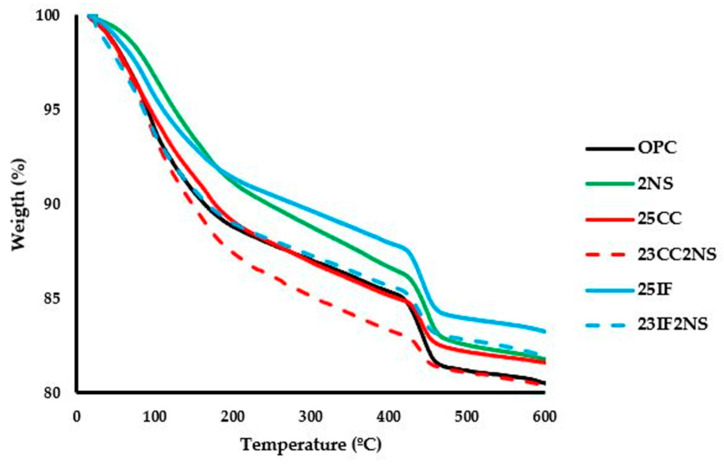
TGA curves, obtained for all pastes at 28 days of hydration.

**Figure 6 materials-16-01837-f006:**
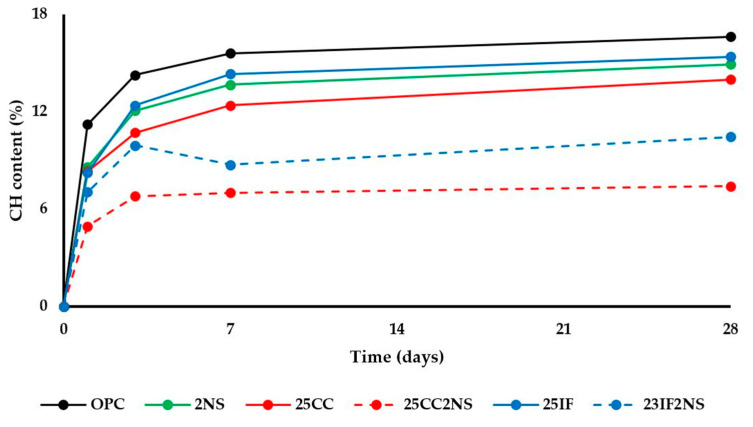
CH content over time for all pastes.

**Figure 7 materials-16-01837-f007:**
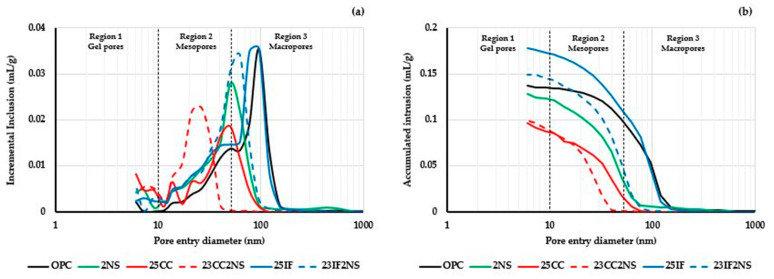
Pore entry diameter distribution of the pastes at 28 days of hydration: (**a**) incremental intrusion and (**b**) accumulated volume. Region 1: Gel pores (<10 nm); Region 2: Mesopores (10–50 nm); Region 3: Macropores (>50 nm).

**Figure 8 materials-16-01837-f008:**
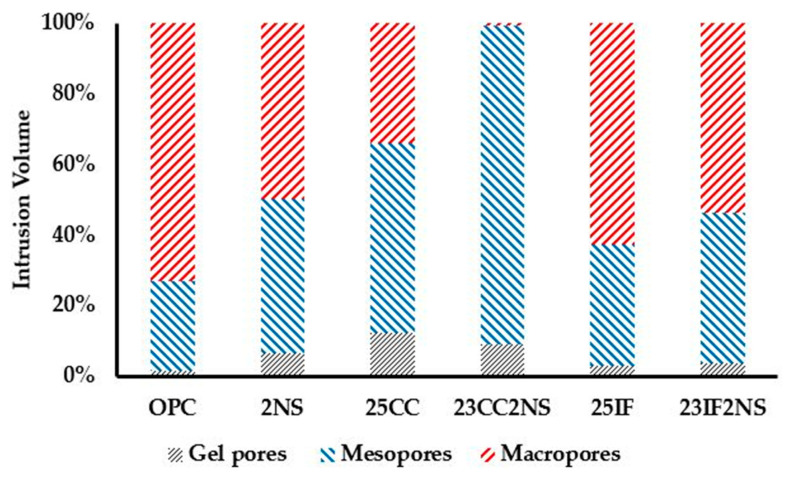
Intrusion volume distribution between regions of gel pores, mesopores, and macropores.

**Figure 9 materials-16-01837-f009:**
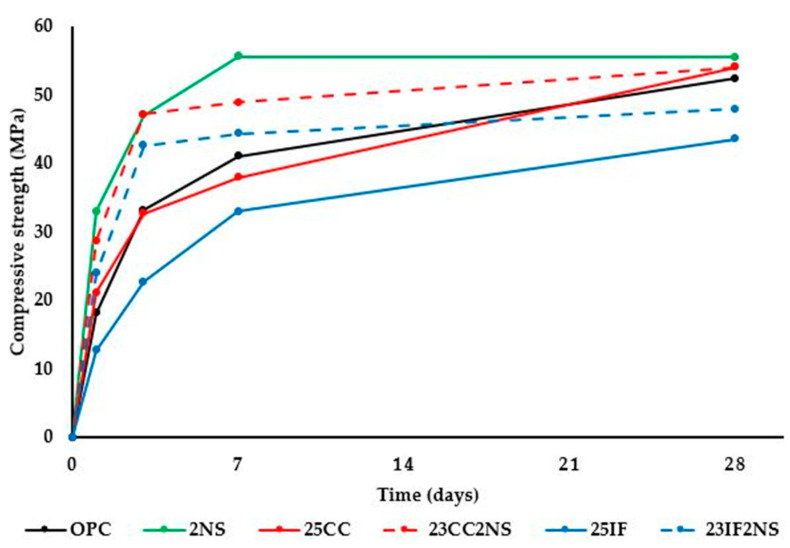
Compressive strength of pastes during hydration up to 28 days.

**Table 1 materials-16-01837-t001:** Chemical and physical properties of OPC, CC, IF, and NS.

Material Property		OPC	CC	IF	NS
Chemical composition (%)	SiO_2_	19.82	57.65	97.83	90.61
	Al_2_O_3_	4.78	34.54	1.48	-
	MgO	5.58	0.26	-	0.1
	Fe_2_O_3_	3.13	3.21	0.57	0.11
	CaO	61.47	-	-	0.02
	Na_2_O	0.11	-	-	2.03
	K_2_O	0.33	1.1	0.26	0.03
	TiO_2_	0.24	1.77	-	0.03
	P_2_O_5_	0.16	-	-	-
	MnO	-	-	-	-
	SO_3_	2.75	-	-	-
	Others	-	-	-	0.9
	Loss on ignition (LOI)	1.47	0.74	0.27	6.19
Specific mass (g/cm^3^)		3.11	2.66	2.69	1.2 *
Specific surface area (m^2^/g)		2.86	17.96	3.013	80 *
Mean particle diameter (μm)		21.65	19.13	23.51	0.022
D10 (μm)		2.19	1.49	1.51	-
D50 (μm)		20.17	14.48	13.81	-
D90 (μm)		42.60	44.25	60.74	-

* Values informed by the manufacturer.

**Table 2 materials-16-01837-t002:** Mass composition of the cement pastes.

Paste	Formulation	OPC (g)	CC (g)	IF (g)	NS Total/Solid Nanosilica (g)	Water (g)	SP (g)/SP (%)	Mini-Slump Diameter (mm)
OPC	100%OPC	1200	-	-	-	480	1.2/0.10%	90
2NS	98%OPC + 2%NS	1176	-	-	80/24	480	9.6/0.80%	92
25CC	75%OPC + 25%CC	900	300	-	-	480	3.6/0.30%	93
23CC2NS	75%OPC + 23%CC + 2%NS	900	276	-	80/24	480	14.1/1.20%	96
25IF	75%OPC + 25%IF	900	-	300	-	480	0.84/0.07%	93
23IF2NS	75%OPC + 23%IF + 2%NS	900	-	276	80/24	480	10.44/0.80%	95

**Table 3 materials-16-01837-t003:** Parameter values obtained from isothermal calorimetry data.

Parameter	OPC	2NS	25CC	23CC2NS	25IF	23IF2NS
Total heat at 24 h (J/g of binder)	237.0	255.3	178.5	197.4	191.0	205.3
Total heat at 72 h (J/g of binder)	350.2	359.1	262.4	294.6	281.1	303.4
Heat flow at main peak (mW/g of clinker)	5.0	7.6	5.2	6.1	5.2	5.7
Main peak maximum point time (h)	12.6	9.1	7.9	8.2	10.6	9.9

**Table 4 materials-16-01837-t004:** Weight loss related to HP index for the pastes at 28 days.

Pastes	OPC	2NS	25CC	23CC2NS	25IF	23IF2NS
Weight loss at Hydrated phases region (50–440 °C)	13.2%	13.0%	13.5%	15.5%	11.1%	12.3%
HP index in relation to OPC	100%	98%	102%	117%	84%	93%

## Data Availability

All data, models, and code generated or used during the study appear in the submitted article.

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
