# Peer review of "Physical and Chemical Effects in Blended Cement Pastes Elaborated with Calcined Clay and Nanosilica"

_materials, 2023, doi:10.3390/ma16051837_

Round 1

Reviewer 1 Report

In this work the influence of calcined clay and nanosilica on hydration processes of blended cement pastes and some properties of hardened material was investigated. An inert filler was also used to better understand the chemical and physical effects of the supplementary cementitious materials.

This manuscript has some errors that need to be corrected. Some statements require additional explanation.

Main remarks:

1) The abbreviations used in cement chemistry to describe chemical compounds (e.g. C2ASH8) and phases (e.g. C-A-S-H) should be explained.

2) The numbering of tables and figures given in the text does not correspond with the numbering in the titles of tables and figures. All numbering and relevant references in the text should be carefully checked and corrected.

3) Part 2.1 Materials:

Lines 125-126, there is: “this CC has metakaolinite content of 50.9% and the calcination process had efficiency of 94%.” - How do the authors know how effective the calcination process was, since the content of metakaolin was calculated theoretically?

Line 122, there is: “dihydroxylation of kaolinite”, it should be “dehydroxylation of kaolinite”

In Fig. 1, the title and unit of the horizontal axis are missing. Some peaks with distinct intensity were not interpreted. This should be corrected.

4) Part 2.2 Past composition and preparation:

Calculating the specific surface area of mixtures based on the results for individual components and giving the value obtained in [m2] is an incorrect approach. These results should be removed.

5) Part 3.3 Thermogravimetric Analysis (TGA):

Lines 327-328, there is: “hydrated phases content was also evaluated at 28 days using the mass loss at the interval 105-400°C” - This temperature range of hydrate dehydration is incorrect. The TG curves show that up to the temperature of 105oC there is a clear loss of mass. Thus,  some part of the bound water is not taken into account during the evaluation of the content of hydrated phases. This can lead to wrong conclusions. Some hydrates, e.g. ettringite, can dehydrate at temperatures below 100oC. In addition, it can be seen from the TG curves that the discussed mass loss ends slightly above 400oC.

6) Part 3.5 Compressive Strength:

Lines 399-400, there is: “Therefore, a limitation in C-S-H content due to undersulfation would lead to lower values in compressive strength at 28 days”.  - On what basis did the authors make such a conclusion?

7) Part 4. Conclusions:

Lines: 418 – 420, there is: “the effect of NS in CH consumption was more significant in paste 23CC2NS than in paste 23IF2NS at all ages, indicating a synergistic effect between NS and CC regarding the pozzolanic reaction” - Comparing the results for these two samples does not confirm the synergistic effect, it only shows that CC has pozzolanic activity compared to inert IF.

Author Response

Thanking you for your pertinent collaborations, we forward the responses to the indicated suggestions. All were carefully appreciated and proved to be of great importance in the development of our work. They are listed below, answered one by one. The authors added texts in the work according to the reviewers' comments. These texts are highlighted in green.

Comment 1) The abbreviations used in cement chemistry to describe chemical compounds (e.g. C2ASH8) and phases (e.g. C-A-S-H) should be explained.

Answer 1: Abbreviations in cement chemistry have been explained as suggested. An example is on page 2 line 60.

Comment 2) The numbering of tables and figures given in the text does not correspond with the numbering in the titles of tables and figures. All numbering and relevant references in the text should be carefully checked and corrected.

Answer 2: The numbering of tables and figures has been carefully revised.

Comment 3) Part 2.1 Materials:

Lines 125-126, there is: “this CC has metakaolinite content of 50.9% and the calcination process had efficiency of 94%.” - How do the authors know how effective the calcination process was, since the content of metakaolin was calculated theoretically?

Answer  3: Based on the reviewer's note, the authors decided to remove the data on the efficiency of calcination. Page 4 line 143.

Comment 4) Line 122, there is: “dihydroxylation of kaolinite”, it should be “dehydroxylation of kaolinite”

Answer  4: Thanks for the note. We corrected the text.

Comment 5) In Fig. 1, the title and unit of the horizontal axis are missing. Some peaks with distinct intensity were not interpreted. This should be corrected.

Answer 5: Thanks for the comment, the figure 1 (on page 5) has been corrected and peaks with distinct intensity have been interpreted.

Comment 6) Part 2.2 Past composition and preparation:

Calculating the specific surface area of mixtures based on the results for individual components and giving the value obtained in [m2] is an incorrect approach. These results should be removed.

Answer 6: Thanks for the observation. Results referring to the specific surface area of mixtures based on individual component results have been excluded from the text. (on page 5, 6 and table 2)

Comment 7) Part 3.3 Thermogravimetric Analysis (TGA):

Lines 327-328, there is: “hydrated phases content was also evaluated at 28 days using the mass loss at the interval 105-400°C” - This temperature range of hydrate dehydration is incorrect. The TG curves show that up to the temperature of 105oC there is a clear loss of mass. Thus,  some part of the bound water is not taken into account during the evaluation of the content of hydrated phases. This can lead to wrong conclusions. Some hydrates, e.g. ettringite, can dehydrate at temperatures below 100oC. In addition, it can be seen from the TG curves that the discussed mass loss ends slightly above 400oC.

Answer 7: The temperature ranges used to analyze the HP index values by the TGA test have been corrected. We have inserted a paragraph to explain this new temperature range, along with new references. The text was adapted and the new data were reanalyzed. Page 10 between line 333 and 353

Comment 8) Part 3.5 Compressive Strength:

Lines 399-400, there is: “Therefore, a limitation in C-S-H content due to undersulfation would lead to lower values in compressive strength at 28 days”.  - On what basis did the authors make such a conclusion?

Answer 8: The authors revised section 3.5 to better present the formulation. An explanation for the undersufation hypothesis was added to the text according to the references found and added in this article. On page 13, between line 407 and 410.

Comment 9) Part 4. Conclusions:

Lines: 418 – 420, there is: “the effect of NS in CH consumption was more significant in paste 23CC2NS than in paste 23IF2NS at all ages, indicating a synergistic effect between NS and CC regarding the pozzolanic reaction” - Comparing the results for these two samples does not confirm the synergistic effect, it only shows that CC has pozzolanic activity compared to inert IF.

Answer 9: The suggestion was analyzed by the authors and the text was corrected as indicated by the reviewer. Page 14, between line 422 and 424.

Reviewer 2 Report

The paper evaluates and analyzes the effects of calcined clay, nanosilica and inert siliceous filler on the cement paste through advanced testing methods. The results and discussions are interesting. It is recommended to publish the paper once the writing can focus on a clear research goal. Other detailed comments are shown below:

(1)        The present title is “Physical and chemical interactions between calcined clay and nanosilica in blended cement pastes”, however, this study mainly reveals and compares the effects of calcined clay, nanosilica and inert siliceous filler, but does not focus on the interaction between calcined clay and nanosilica. It is suggested to revise the title to better reflect the main content of the paper.

(2)        L48 Portlandite (CH). CH is set to be the abbreviation of Portlandite. It is noticed that CH occurs in many places such as “Its physical properties promote the acceleration of the hydration reaction, mainly due to a potentialized nucleation effect, and chemically it reacts with CH to form addition C-S-H as early as 1 day in the hydration process.”, “The pozzolanic reaction happened at both and early ages due to the combination of the materials, noted by lower CH content and higher C-A-S-H content, when compared to OPC pastes.” and “Since the CH content was lower in these conditions, the only explanation is pozzolanic activity.” Is CH the shortened term of Portlandite throughout the paper?

(3)        L65 “The pozzolanic reaction happened at both and early ages due to the combination of the materials.” What does “both” mean?

(4)        L68 “Considering the similarities between metakaolin and calcined kaolinitic clays” What is the difference between metakaolin and calcined kaolinitic clay?

(5)        L94 “The grinding time was adjusted by taking the granulometry of OPC as a parameter, and the final milling time for CC and IF was 4 and 8 hours, respectively.” It is recommended to provide more information about how to determine the milling time.

(6)        L147 “This consistency was achieved by using superplasticizer admixture.” Since SP is a component of mixes in this study, it is suggested to provide the information of SP such as the kind, the physical properties and the working mechanism of SP.

(7)        L154 “Table 1” should be Table 2.

(8)        L164 “The pastes were prepared in a climatized room with temperature at 23±1 °C.” Why set 23±1 °C? According to what specifications?

(9)        L207 “A few parameters were obtained by analysis of the curves and are presented in Table 3.” Does this statement match the data in Table 3?

(10)     L244 “OPC paste and 25IF paste, which had the aluminate peak happening later had similar total surface area, as shown in Table 2, close to 3500m².” Does this statement match the data in Table 2?

(11)     L324 “This behavior was repeated in the following ages, although the difference decreased. This result shows that there is an synergistic interaction between CC and NS supporting previously reported results [57,63].” It is suggested to briefly introduce "previously reported results [57,63]". Similarly, L390 “The overall effect of CC, in both binary and ternary mixture, was lower than expected, considering previous works [67,68], particularly in the case of paste 23CC2NS.” It is suggested to briefly introduce " previous works [67,68]".

(12)     L327 “Furthermore, hydrated phases content was also evaluated at 28 days using the mass loss at the interval 105-400°C [49].” What does "hydrated phases content" mean? It is suggested to provide a brief explanation.

(13)     L341 “In this case, the hydrated phases produced by OPC hydration and pozzolanic reaction surpasses the hydrated phases of OPC paste, even with a smaller amount of clicker.” What does "even with a small amount of clicker" mean?

(14)     L353 “Figure 6(a) shows that there was refinement of the pore structure in pastes containing CC and/or NS” and L358 “Figure 6(b) shows the accumulated intrusion, and the maximum value is related to the total porosity in the sample.” Please check the figure number.

(15)     L374 “Compressive strength values were obtained for the pastes at 1, 3, 7 and 28 days of hydration and are presented in graph form in Figure 8.” Please check the figure number.

Author Response

Thanking you for your pertinent collaborations, we forward the responses to the indicated suggestions. All were carefully appreciated and proved to be of great importance in the development of our work. They are listed below, answered one by one.The authors added texts in the work according to the reviewers' comments. These texts are highlighted in green.

Comment 1) The present title is “Physical and chemical interactions between calcined clay and nanosilica in blended cement pastes”, however, this study mainly reveals and compares the effects of calcined clay, nanosilica and inert siliceous filler, but does not focus on the interaction between calcined clay and nanosilica. It is suggested to revise the title to better reflect the main content of the paper.

Answer 1: The authors accepted the reviewer's suggestion, and the title of the article became: Physical and chemical effect in blended cement pastes elaborated with calcined clay and nanosilica.

Comment 2)        L48 Portlandite (CH). CH is set to be the abbreviation of Portlandite. It is noticed that CH occurs in many places such as “Its physical properties promote the acceleration of the hydration reaction, mainly due to a potentialized nucleation effect, and chemically it reacts with CH to form addition C-S-H as early as 1 day in the hydration process.”, “The pozzolanic reaction happened at both and early ages due to the combination of the materials, noted by lower CH content and higher C-A-S-H content, when compared to OPC pastes.” and “Since the CH content was lower in these conditions, the only explanation is pozzolanic activity.” Is CH the shortened term of Portlandite throughout the paper?

Answer 2: Yes. The abbreviation CH refers to Portlandite throughout the article.

Comment 3)        L65 “The pozzolanic reaction happened at both and early ages due to the combination of the materials.” What does “both” mean?

Answer 3: The term both has not been correctly applied. We explain that reactions take place at early and later ages when materials are combined. Appreciate

Comment 4)        L68 “Considering the similarities between metakaolin and calcined kaolinitic clays” What is the difference between metakaolin and calcined kaolinitic clay?

Answer 4: The suggestion was analyzed and a part was inserted in the text that justifies the difference between metakaolin and calcined calinitic clay, as can be seen, page 2 line 53

Comment 5)        L94 “The grinding time was adjusted by taking the granulometry of OPC as a parameter, and the final milling time for CC and IF was 4 and 8 hours, respectively.” It is recommended to provide more information about how to determine the milling time.

Answer 5: The suggestion was analyzed and more information about the process of determining the grinding time was inserted in the text. Page 3, line 103 to 105

Comment 6)        L147 “This consistency was achieved by using superplasticizer admixture.” Since SP is a component of mixes in this study, it is suggested to provide the information of SP such as the kind, the physical properties and the working mechanism of SP.

Answer 6: The suggestion was analyzed and Information about the superplasticizer was added to the text. Page 2 line 94 to 97

Comment 7)        L154 “Table 1” should be Table 2.

Answer 7: The observation was checked and it was found that the correct one is table 2

Comment 8)        L164 “The pastes were prepared in a climatized room with temperature at 23±1 °C.” Why set 23±1 °C? According to what specifications?

Answer 8: The pastes were prepared in accordance with the Brazilian standard ABNT 7215:2019, and this norm determines that the temperature of the acclimatized room must be 23±1°C. The reference of the norm was inserted on page 6 line 179.

Comment 9)        L207 “A few parameters were obtained by analysis of the curves and are presented in Table 3.” Does this statement match the data in Table 3?

Answer 9: Yes. Its parameters presented in table 3 are obtained through analysis of figure 3. To make it clearer we changed the text on page 7 line 222.

Comment  10)     L244 “OPC paste and 25IF paste, which had the aluminate peak happening later had similar total surface area, as shown in Table 2, close to 3500m².” Does this statement match the data in Table 2?

Answer 10: This data was removed from the text at the suggestion of another reviewer.

Comment 11)     L324 “This behavior was repeated in the following ages, although the difference decreased. This result shows that there is an synergistic interaction between CC and NS supporting previously reported results [57,63].” It is suggested to briefly introduce "previously reported results [57,63]". Similarly, L390 “The overall effect of CC, in both binary and ternary mixture, was lower than expected, considering previous works [67,68], particularly in the case of paste 23CC2NS.” It is suggested to briefly introduce " previous works [67,68]".

Answer 11: The suggestion was analysed and a text was added explaining both the references and the references, page 11 line 348

Comment 12)     L327 “Furthermore, hydrated phases content was also evaluated at 28 days using the mass loss at the interval 105-400°C [49].” What does "hydrated phases content" mean? It is suggested to provide a brief explanation.

Answer 12: The part about the temperature range has been changed following suggestions from another reviewer and the text has been modified as noted on page 10 between lines 333 and 341.

Comment 13)     L341 “In this case, the hydrated phases produced by OPC hydration and pozzolanic reaction surpasses the hydrated phases of OPC paste, even with a smaller amount of clicker.” What does "even with a small amount of clicker" mean?

Answer 13: The suggestion was analysed, and the text was changed in order to make the understanding on this subject clearer, as can be seen on page 11 line 349

Comment 14)     L353 “Figure 6(a) shows that there was refinement of the pore structure in pastes containing CC and/or NS” and L358 “Figure 6(b) shows the accumulated intrusion, and the maximum value is related to the total porosity in the sample.” Please check the figure number.

Answer 14: All figure numbers and tables in the text have been carefully checked.

Comment 15)     L374 “Compressive strength values were obtained for the pastes at 1, 3, 7 and 28 days of hydration and are presented in graph form in Figure 8.” Please check the figure number.

Answer 15: All figure numbers and tables in the text have been carefully checked.

Reviewer 3 Report

Title: “Physical and chemichal interactions between calcinated clay and nanosilica in blended cement pastes”

In line 50 it is worth mentioning that in the case of SCM’s later ages of 60 or 90 days might be of interest, given the late reaction of the SCM.

In line 91 it is mentioned that ideal calcination temperatures were chosen according to literature, however, TGA analysis could also evidence the ideal calcination temperature for CC. While in line 124 is indeed mentioned that TGA was performed for CC, the TGA curve of CC could be very informative for readers to better illustrate the mass loss behavior depicted here.

Title in 2.2 should be “Paste composition and preparation”

In line 227, the early hydration is attributed mainly to the nucleation effect, however, in the case of pozzolanic reaction, CH consumption to precipitate hydrated products also plays a key role during the accelerating stage and could be considered as well, a behavior explained well in line 48 and 304.

In line 242, “… and more sulfate *gets* adsorbed by it…”

Performing a SEM analysis is strongly recommended to visualize the hydrated products and pore structure mentioned in section 3.2 and section 3.4.

Recommended literature cites regarding SEM analysis, XRD analysis, microstructure and Specific Surface on SCM’s and blended pastes:

- Castillo, D., Cruz, J. C., Trejo-Arroyo, D. L., Muzquiz, E. M., Zarhri, Z., Gurrola, M. P., & Vega-Azamar, R. E. (2022). Characterization of poultry litter ashes as a supplementary cementitious material. Case Studies in Construction Materials, 17, e01278. https://doi.org/10.1016/j.cscm.2022.e01278

- Ram, K., Serdar, M., Londono-Zuluaga, D., & Scrivener, K. (2022). The effect of pore microstructure on strength and chloride ingress in blended cement based on low kaolin clay. Case Studies in Construction Materials, 17, e01242. https://doi.org/10.1016/j.cscm.2022.e01242

In line 269 an interesting way to determine pozzolanic reaction is described by CH consumption depicted in the diffractograms. Nonetheless, to confirm these this reaction two different quick tests should also be considered: the Fratinni test and Chapelle test, which allow for determination of pozzolanic activity by CH consumption. It is strongly recommended to perform these tests.

In section 3.3 TGA curves for blended cement pastes are shown, while in previous sections the TGA curve of CC should be added as well.

In line 317, Fratinni and Chapelle test could reinforce this statement, given these tests are efficient, reproducible and widespread among blended cement pastes studies.

Author Response

Thanking you for your pertinent collaborations, we forward the responses to the indicated suggestions. All were carefully appreciated and proved to be of great importance in the development of our work. They are listed below, answered one by one.The authors added texts in the work according to the reviewers' comments. These texts are highlighted in green.

Comment 1) In line 50 it is worth mentioning that in the case of SCM’s later ages of 60 or 90 days might be of interest, given the late reaction of the SCM.

Answer 1 : The observation was analyzed and inserted in the text about the ages after the greater effect of WC in more advanced ages, as can be seen in line 66 page 2.

Comment 2) In line 91 it is mentioned that ideal calcination temperatures were chosen according to literature, however, TGA analysis could also evidence the ideal calcination temperature for CC. While in line 124 is indeed mentioned that TGA was performed for CC, the TGA curve of CC could be very informative for readers to better illustrate the mass loss behavior depicted here.

Answer 2: The suggestion was analyzed and the TGA curve of the clay in natura was inserted in the article to bring more information to the reader. On line 4 page 138.

Comment 3) Title in 2.2 should be “Paste composition and preparation”

Answer 3: We made the modification.

Comment 4) In line 227, the early hydration is attributed mainly to the nucleation effect, however, in the case of pozzolanic reaction, CH consumption to precipitate hydrated products also plays a key role during the accelerating stage and could be considered as well, a behavior explained well in line 48 and 304.

Answer 4: The observation was analyzed and the text was changed in order to better explain the behavior. Page 7 line 236

Comment 5) In line 242, “… and more sulfate *gets* adsorbed by it…”

Answer 5: We made the modification.

Comment 6) Performing a SEM analysis is strongly recommended to visualize the hydrated products and pore structure mentioned in section 3.2 and section 3.4.

Answer 6: The SEM test is very interesting for future research, but due to the dead line to send the corrections we will not have time to insert it in this work.

Comment 7) Recommended literature cites regarding SEM analysis, XRD analysis, microstructure and Specific Surface on SCM’s and blended pastes:

- Castillo, D., Cruz, J. C., Trejo-Arroyo, D. L., Muzquiz, E. M., Zarhri, Z., Gurrola, M. P., & Vega-Azamar, R. E. (2022). Characterization of poultry litter ashes as a supplementary cementitious material. Case Studies in Construction Materials, 17, e01278. https://doi.org/10.1016/j.cscm.2022.e01278

- Ram, K., Serdar, M., Londono-Zuluaga, D., & Scrivener, K. (2022). The effect of pore microstructure on strength and chloride ingress in blended cement based on low kaolin clay. Case Studies in Construction Materials, 17, e01242. https://doi.org/10.1016/j.cscm.2022.e01242

Answer 7: The articles were analyzed and added as references in this article.

Comment 8) In line 269 an interesting way to determine pozzolanic reaction is described by CH consumption depicted in the diffractograms. Nonetheless, to confirm these this reaction two different quick tests should also be considered: the Fratinni test and Chapelle test, which allow for determination of pozzolanic activity by CH consumption. It is strongly recommended to perform these tests.

Answer 8: The Fratinni test and Chapelle test are very interesting for future research, but due to the dead line to send the corrections we will not have time to insert it in this article.

Comment 9) In section 3.3 TGA curves for blended cement pastes are shown, while in previous sections the TGA curve of CC should be added as well.

Answer 9: The TGA curve of clay in natura was inserted in the article as recommended, page 4 line 138

Comment 10) In line 317, Fratinni and Chapelle test could reinforce this statement, given these tests are efficient, reproducible and widespread among blended cement pastes studies.

Answer 10: The Fratine test and Chapelle test are very interesting for future research, but due to the dead line to send the corrections we will not have time to insert it in this work.

Reviewer 4 Report

The work presented in this manuscript is very detailed and providing very interesting results when replacing 25% of clinker with 23% of calcined clay and 2% nanosilica.

However, there are certain corrections required. The Materials and Methods section needs to be reduced; thus, it is common to place all the results (even the ones for the characterization of the starting materials).

XRPD results are not explained as they should. Please make sure to cover all the peaks adequately (only the ones in Fig 1).

Author Response

Thanking you for your pertinent collaborations, we forward the responses to the indicated suggestions. All were carefully appreciated and proved to be of great importance in the development of our work. They are listed below, answered one by one. The authors added texts in the work according to the reviewers' comments. These texts are highlighted in green.

Comment 1) (...) The Materials and Methods section needs to be reduced; thus, it is common to place all the results (even the ones for the characterization of the starting materials).

Answer 1:Thanks for the suggestion, we carefully reviewed the section and found the information necessary for the reader's good understanding.

Comment 2) XRPD results are not explained as they should. Please make sure to cover all the peaks adequately (only the ones in Fig 1).

Answer 2: The figure 1 was corrected and the peaks were indicated as suggested.

Reviewer 5 Report

The authors aimed to develop different types of cement. For this purpose, silica filler fillers have substituted calcined clay from clinker. In addition to these mixtures, nano silica was added. Compressive strength, Isothermal calorimetry, Thermogravimetry (TG/DTG), X-ray diffraction (XDR) and Mercury intrusion Porosimetry (MIP) tests were performed on these samples.

The following shortcomings have been identified in this manuscript;

Numerical data should be included in the abstract.

In table 1, it is given as Sp (%), the % sign in the table should be removed

-In addition to references 1-7 in the introduction, the following resources can be added. this part could be more scientific and up to date

10.1061/(ASCE)MT.1943-5533.0004425.

10.1016/j.jobe.2022.104849

In general, this article can be used.

10.1016/j.conbuildmat.2017.10.041

Author Response

Thanking you for your pertinent collaborations, we forward the responses to the indicated suggestions. All were carefully appreciated and proved to be of great importance in the development of our work. They are listed below, answered one by one. The authors added texts in the work according to the reviewers' comments. These texts are highlighted in green.

Comment 1) Numerical data should be included in the abstract.

Answer 1: We thank you for the suggestion and insert numeric data in the abstract, as from line 31.

Comment 2) In table 1, it is given as Sp (%), the % sign in the table should be removed

Answer 2: We removed the column with the percentages from Table 1 and inserted the information together with the mass of superplasticizer.

Comment 3) In addition to references 1-7 in the introduction, the following resources can be added. this part could be more scientific and up to date

10.1061/(ASCE)MT.1943-5533.0004425.

10.1016/j.jobe.2022.104849

In general, this article can be used.

10.1016/j.conbuildmat.2017.10.041

Answer 3: Thanks for the suggestion. They are interesting references and have been added to the text.

Reviewer 6 Report

materials-2181416

Article title:

Physical and chemical interactions between calcined clay and nanosilica in blended cement pastes

Comments:

The main aim of this work was focused on the effect of CC and nanosilica. Great efforts have been performed with sufficient data. Some small suggestions were raised for consideration.

1. Please provide some discussions in an ‘Introduction’ part or ‘Conclusion’ part related to embodied carbon from getting that calcined clay. LCA or comparison of CO2 emission in total may need to be discussed.

2. Figure 3, please expand or re-adjust those XRD patterns e.g. as Fig 3a, 3b… . It is better to explain in larger figs.    

Author Response

Thanking you for your pertinent collaborations, we forward the responses to the indicated suggestions. All were carefully appreciated and proved to be of great importance in the development of our work. They are listed below, answered one by one. The authors added texts in the work according to the reviewers' comments. These texts are highlighted in green.

Comment 1) Please provide some discussions in an ‘Introduction’ part or ‘Conclusion’ part related to embodied carbon from getting that calcined clay. LCA or comparison of CO2 emission in total may need to be discussed.

Answer 1: Thanks for the suggestion. This subject is very important and was inserted in the introduction page 1 line 44 to 50

Comment 2). Figure 3, please expand or readjust those XRD patterns e.g. as Fig 3a, 3b… . It is better to explain in larger figs.

Answer 2: The figure was corrected and the peaks that were not indicated were explained.

Round 2

Reviewer 1 Report

The manuscript was corrected according to my previous remarks, but not completely. I also found a few mistakes.

* The authors have included an explanation of the abbreviations, as follows: “… C-A-S-H (hydrated calcium aluminosilicate), and aluminate hydrates, such as C2ASH8 (stratlingite) and C4AH12” (page 2, lines 60-61). To be precise, C2ASH8 also belongs to the group of hydrated aluminosilicates.

* In my previous review, I pointed out  that phrase “dihydroxylation of kaolinite” is wrong and it should be: “dehydroxylation of kaolinite”. The authors asserted in their answer that this has been changed, but it is not. In addition, "dihydroxylation" appeared in several other places in this work. It should be: "dehydration" or "dehydroxylation" depending on chemical compound or phase and temperature.

* page 7, lines 222-223, there is: “Figure 3(a) shows the heat flow curve, regularized by clinker content in the pastes, while Figure 3(b) shows their overall heat of hydration”. Figure 3 is different: Fig. 3(a) shows overall heat of hydration and Fig. 3(b) shows the heat flow curves.

* The data in Table 3 and Fig. 3 should be checked. The data in Table 3 is not consistent with the data in Fig. 3. For example: in the case of the Heat flow at main peak, the highest value is for the sample 23CC2NS (according to Table 3) while in the Fig. 3 the highest value is for 2NS. In the case of 25CC, the value visible in Fig.3 is below 6, while in the Table 3 it is 7.0. Etc.

* There is in Table 7: “105-400oC”, while other temperature range is in the text: 50-440oC (page 10, line 336).

* The data in Table 7 should be checked. For example: in the case of Weight loss at Hydrated phases region, the value for 2NS is higher than for OPC, while one can see in Fig.5 that this mass loss is lower.

Author Response

see file

Reviewer 2 Report

The authors have responded the comments. It is recommended to check the writing and make the presentation clearer.

Author Response

see file
